# Community Perception of Animal-Based Urban Agriculture within City Greenspaces of the Global North: A Survey of Residents near Cornwall Park, New Zealand

**Shannon Davis * and Guanyu Chen** 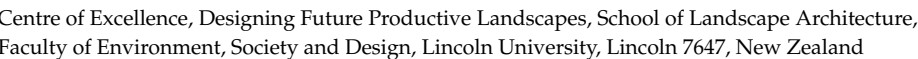

Centre of Excellence, Designing Future Productive Landscapes, School of Landscape Architecture, Faculty of Environment, Society and Design, Lincoln University, Lincoln 7647, New Zealand
* Correspondence: shannon.davis@lincoln.ac.nz; Tel.: +64-3-423-0473

**Abstract:** The ability of cities worldwide to feed themselves is of increasing concern to Urban Planning and Design professions. In recent years, interest in reintegrating agricultural production back into cities of the Global North has grown, particularly with regard to plant-based urban agriculture. Research focused on animal-based urban agriculture however has been notably absent from the literature and case studies set within cities of the Global North. This study aims to contribute to this emerging area of research and seeks to better understand the enablers and barriers to integrating grazing animals within urban greenspace from a 'neighbor' perspective. This paper presents survey responses from residents living around Cornwall Park, an urban greenspace in Aotearoa New Zealand's largest city, Tāmaki Makaurau Auckland, that integrates a working sheep and beef farm as part of the 172 ha urban greenspace. Findings revealed that the grazing animals were highly valued by the neighboring community with 99% of respondents feeling 'positive' towards the inclusion of grazing animals as part of the public park. Our findings have implications for cities of the Global North considering the reintegration of animal-based urban agriculture, providing support for decision-making when defining policies for enabling animal-based agriculture within public greenspace.

**Keywords:** urban agriculture; urban greenspace; animal agriculture; grazing lands

## 1. Introduction

In recent years, urban agriculture has gained increasing attention from the Urban Planning and Design professions primarily in response to a growing movement towards increasing the resilience of our cities and urban settlements [1–5]. As discussed by Hanna and Wallace [6], urban issues such as food insecurity, climate change and disasters, urban decay, the promotion of 'wellbeing' and the COVID-19 pandemic have all motivated the rising interest in agriculture within urban spaces. As a basic human need for survival, access to food and agricultural products is considered an essential component of an urban system, but during the past 100 years, cities of the Global North have become increasingly reliant on long global food supply chains, removing productive landscapes as an acceptable and common landuse from urban limits, and therefore spatially disconnecting themselves from food production. Technology, transport and logistical organization has developed to allow this spatial disconnect, alongside changes in urban lifestyle preference, employment opportunities, land prices, and the perception of competing and non-compatible urban land use [7–12]. Productive agricultural landscapes as a result, have been 'pushed' further and further away from urban settings, and this landuse divide has been supported by contemporary urban planning practices and policies [6,13–16]. This spatial divide and reliance on agricultural products being produced in 'rural' zones separate to the 'urban' zones has a direct effect on the overall resilience of cities.

The recent rise however that has been well-documented in relation to growing interest and activity in urban agriculture has primarily focused on plant-based systems. The 2019 study by Graefe, Buerkert and Schlecht [17], found that just 2% of scholarly studies on urban and peri-urban agriculture dealt with animal husbandry, with most of these based on case studies located in the Global South.

Animal-based urban agriculture within cities of the Global North has not been widely considered alongside the recent rise and interest in re-establishing agricultural production, which has focused primarily on plant-based practices such as community gardens, allotments, and food forests, alongside more intensive practices of rooftop and indoor plant-based growing initiatives. Although once providing many cities of the Global North with vital waste management and transportation, in addition to an important protein and fiber supply, animal-based agriculture within cities of the Global North has, for the past 100 years, been actively planned out of city limits, and in many cases out of the peri-urban hinterland also [14,18]. Overcrowded slaughterhouses and stockyards, disease, and animal pollution, alongside growing urban populations and increasing demand for urban land all contributed to the planned movement of animals out of many cities of the Global North during the 19th century. Across the Global North local planning regulations were authorized that saw the mass removal of animals from cities [14,16]. Blecha and Leitner [13] state "Since the mid-nineteenth century, poultry and livestock animals have increasingly been seen as out of place in, and excluded from, modern . . . cities" (p. 86). City residents of the Global North have on the whole, come to understand productive animals only as farm animals, out of place by definition in an urban setting" [13]. There were some well-known examples of city greenspaces which did continue to include pastoral landscapes with grazing animals into the 1900s. Central Park in New York, for example, has an area Sheep Meadow where sheep grazed from early after the park's establishment in 1864 through until 1934. After that time the sheep were moved to Prospect Park in Brooklyn which housed sheep until the 1940's [14], However, animals had been removed from the vast majority of cities by the turn of the 20th century.

Investigations of animal-based urban agriculture have centered predominantly within cities of the Global South, looking at issues of both subsistence-based and market-orientated production, addressing issues such as poverty reduction, food supplementation, and livelihood strategies [17]. As interest in urban agriculture has risen, so too has the interest in urban animal keeping within cities of the Global North, with a diversification in farming and keeping practices observed. For many urban livestock keepers, the focus has generally been on egg, milk or honey production, but today, according to Blecha [19], urban residents are increasingly choosing to keep animals for meat (p. 34). In many cities, therefore, Urban Planners and Designers are attempting to update plans and policies to reflect changing land uses and activities, which include the production and sale of agricultural products and the keeping of urban livestock [20].

In the 2016 study by Specht et al. [21], investigating socially acceptable urban agriculture businesses, low acceptance of urban animal agricultural products was illustrated, with two main reasons being provided for these objections. First, individuals suspected that their quality of life would decrease due to intensified odors and noise; and secondly, urban environments were perceived by urban residents as "unnatural" spaces for raising livestock. The fact that these animal products are met with low consumer acceptance (in this case with 70% of respondents rejecting animal products from urban areas) highlights key barriers for those who wish to establish animal-based urban agriculture.

As part of understanding these barriers to re-integrating livestock for food production back into cities of the Global North, this research focuses on the community perception of including grazing animals within an urban greenspace to better understand, from a neighbor/community perspective the attitudes and experiences of urban residents with a working urban farm.

This research contributes to a better understanding of the enablers and barriers to re-integrating grazing animals within cities of the Global North. Focused on community

perceptions towards the inclusion of a sheep and cattle farm within an urban park, this research surveyed residents living near Cornwall Park to better understand their attitudes and experiences with regard to the animals, therefore identifying a range of enablers and barriers to including animal-based urban agriculture within urban parks.

## 2. Method

The case study method was adopted to explore the attitudes of neighbor communities to animal-based urban agriculture. The case study method has a long and well-established history in landscape architecture providing a research approach that brings to light projects that serve as exemplars from which to better understand design processes, concepts and outcomes [22] (p. 1). Cornwall Park was selected as the case study site due to the integration of a working sheep and beef farm within the urban greenspace context that is spatially surrounded by primarily residential land use. The case study provided an opportunity to investigate community perceptions of an established animal-based urban agriculture system operating within an urban setting. As defined by Francis [22], a "case study is a well-documented and systematic examination of the process, decision-making and outcomes of a project, which is undertaken for the purpose of informing future practice, policy, theory, and/or education" (p. 16).

Cornwall Park in New Zealand's largest city of Tāmaki Makaurau Auckland offers a case study site from the Global North context where animal-based agriculture is integrated into the 172 ha urban greenspace (Figure 1).

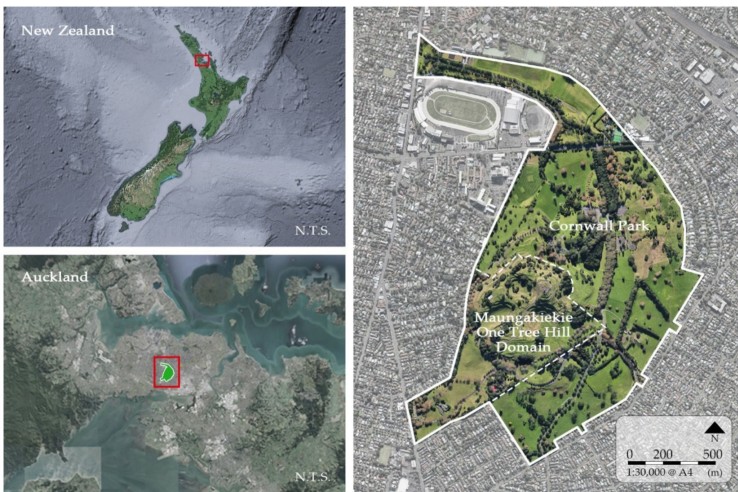

**Figure 1.** Aerial Photograph of Cornwall Park (Adapted from Google Earth, Image © 2022 Maxar Technologies, Image © 2022 CNES/Airbus, Map data SIO, NOAA, U.S. Navy, NGA, GEBCO).

Surrounded by continuous urban land use, the public park integrates a working sheep and beef farm alongside the more conventional Park features such as recreational walking tracks; café, restaurant and educational hub; fitness circuit; playground; native, amenity and ornamental plantings; as well as sites of archaeological and historical significance; and vast lawns. The park was originally designed by Landscape Architect Austin Strong in the early 1900s who was commissioned by the land owner Sir John Logan Campbell in 1901 to create a park for all the people of New Zealand Including grazing lands of 75 hectares (approx.) within Cornwall Park and 44 hectares (approx.) within the neighboring Maungakiekie/One Tree Hill Domain (Figure 2), the Park runs both a stud herd of Simmental cattle, and sheep for lamb meat production and wool/pelts, including Perendale, Texel Cross and Gotland flocks (Figure 3).

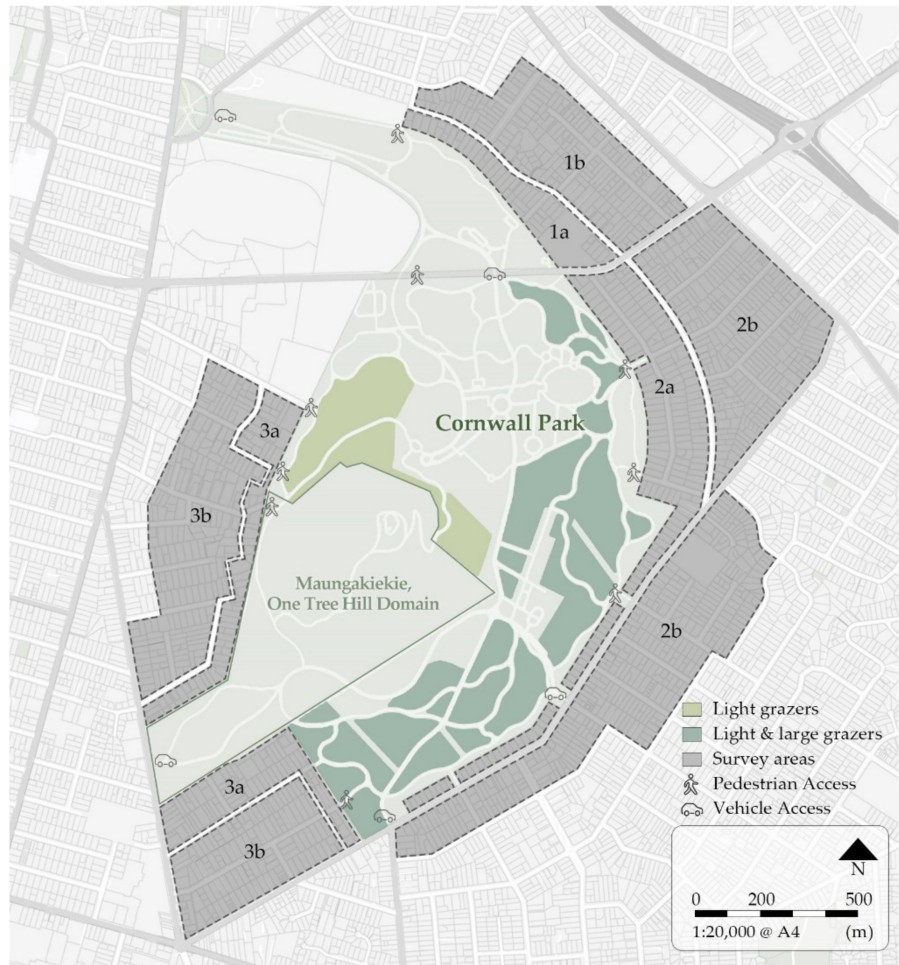

**Figure 2.** Grazed pastureland at Cornwall Park. (Based on pastureland spatial information from Boffa Miskell Ltd. & Nelson Byrd Woltz Landscape Architects, 2014, [23] p. 155).

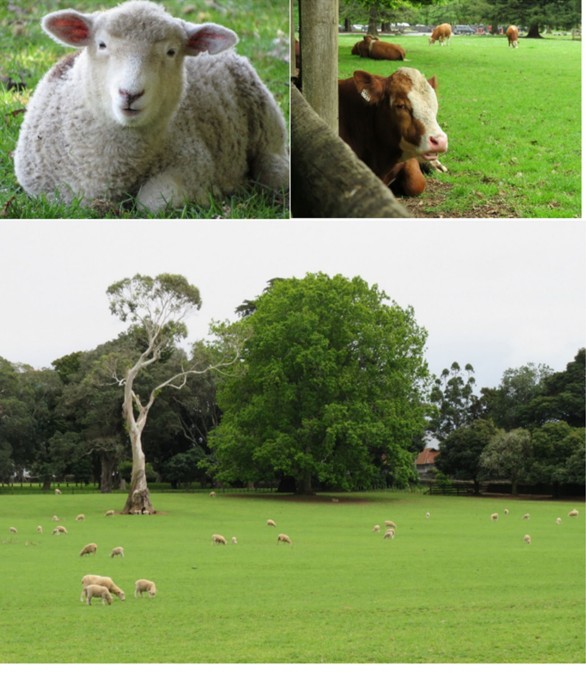

**Figure 3.** Sheep and cattle grazing within the farm at Cornwall Park (Author, 2021).

### *2.1. Participants*

### 2.1.1. Participant Characteristics

The data were collected via a survey distributed to households in the neighborhoods bordering Cornwall Park (Figure 4). In total 400 surveys were delivered to letterboxes of households surrounding Cornwall Park, comprising both land and homeowners, and residential tenants. One participant per household (over the age of 18) was invited to complete the survey.

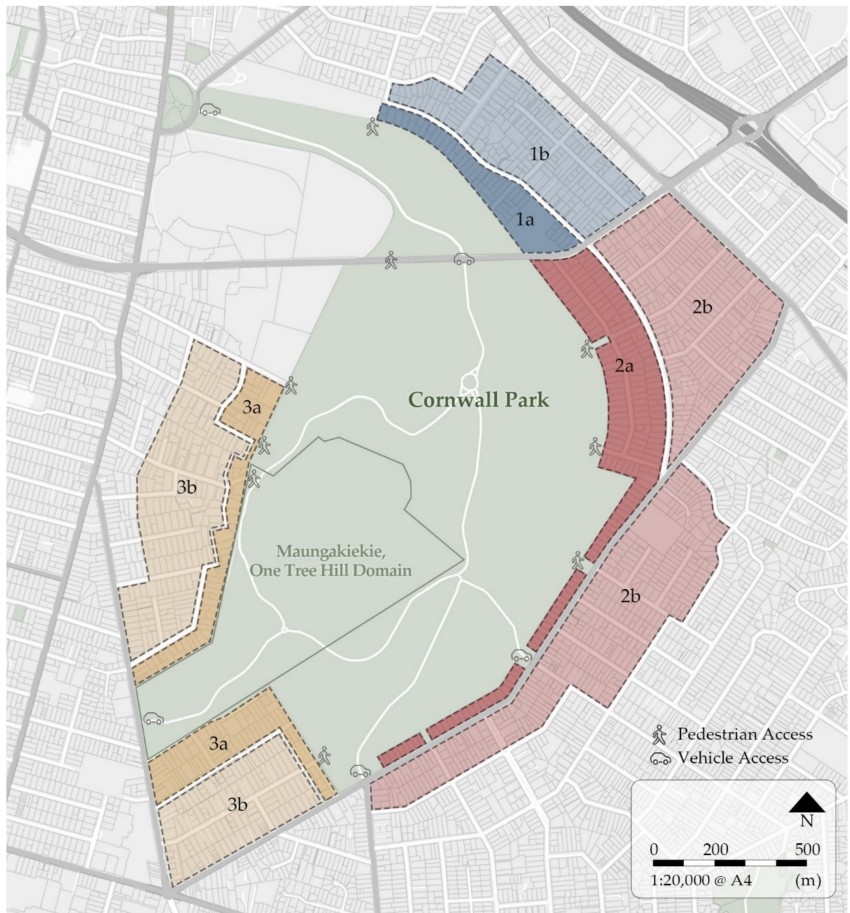

**Figure 4.** Survey Zones Map.

Surveys were grouped into six zones to allow identification of any differences between response based on proximity to the areas of the park used for grazing (Figure 4).

### 2.1.2. Sampling Procedure

Surveys were distributed to households via a letterbox drop. Criteria for leaving a survey included being located within the study area, and the ability to leave and then collect a completed survey (some properties did not have a letterbox, and some, who had letterboxes, were only assessable by key meaning a completed survey could not be collected). Participants were given two days to complete the survey and return it to their letterbox for collection. Participants were asked to place a sticker on their letterbox to indicate a completed survey for collection.

### 2.1.3. Sample Size

The survey yielded a total of 83 responses, of which, 73 were accepted, and 10 were rejected due to incomplete consent forms. The overall response rate is 18%. Statistically, the sampling has a margin of error of 10% at a 95% level of confidence, which is acceptable for statistical analysis.

The summary statistics for the sample are presented in Table 1.

**Table 1.** Socio-demographic profile of respondents.

|  |  | Percentage (%) |
|---|---|---|
| Age range | 18–35 | 6 |
|  | 36–65 | 55 |
|  | 66+ | 40 |
| Household type | One person | 16 |
|  | Couple | 43 |
|  | House share | 4 |
|  | Family (incl. children) | 34 |
|  | Family (excl. children) | 3 |
| Ownership | House & land owner | 69 |
|  | House owner | 16 |
|  | Tenant | 15 |
| Proximity to Park | Home directly borders Cornwall Park | 41 |
|  | Home within a block that directly borders Cornwall Park | 44 |
|  | Home within a block that does not directly border Cornwall Park | 10 |
|  | Unknown | 6 |

### 2.2. Sample Characteristics and Baseline

The survey focused on elements that were important to understanding the perceptions and experiences of neighbors in relations to the grazing animals at Cornwall Park. Issues were also selected that had potential to inform urban agriculture policy. Understanding how and how often residents used the Park was important in establishing a baseline understanding of engagement with the park facilities (Table 2).

**Table 2.** Park visitation profile of respondents.

|  |  | Percentage (%) |
|---|---|---|
| Park visitation frequency | Everyday | 27 |
|  | Once a week or more | 60 |
|  | Once a month or more | 12 |
|  | Less than once a month | 0 |
| Visitation motivation | Recreation | 73 |
|  | Dog walking | 33 |
|  | Commuting | 8 |
|  | Exercise | 92 |
|  | Using Park facilities | 27 |
|  | Other | 12 |

After establishing the base-line data of respondent profiles and general engagement with the park, the survey asked respondents to rate the level of importance of the grazing animals (Table 3).

**Table 3.** Attitudes towards grazing animals in Cornwall Park.

|  |  | Percentage (%) |
|---|---|---|
| Attitude towards grazing animals at Cornwall Park | Extremely positive | 89 |
|  | Mostly positive | 10 |
|  | Neutral | 1 |
|  | Mostly negative | 0 |
|  | Extremely negative | 0 |

*2.3. Procedure*

Data Collection Methods

Respondents filled in a paper-based survey consisting of ten 'tick-box' questions (to establish the base-line) and then four short answer questions. Participants were told that the survey would take 10–15 min, with one resident per household, over the age of 18, able to participate.

*2.4. Data Analysis*

The quantitative data collected through the ten multi-choice questions were analyzed using SPSS 26. A Spearman's rank-order correlation was run to determine the correlation significance between the residents' attitudes towards grazing animals and the other ordinal variables, including socio-demographical attributes (i.e., age, ownership, and proximity) and park visitation attributes (i.e., the importance for residing decision, and use frequency). The relationships between the residents' attitudes and the categorical variable (i.e., household type) and the binary variable (i.e., park use) were examined by using the Pearson Chi-Square test.

Participant responses to the four short answer questions were first transcribed and then coded with a line-by-line coding approach to draw all the identifiable ideas indicated by the respondents from the texts. Overall, 120 codes were created. These codes were then categorized and counted to reflect the key opinions reported by the respondents. The correlation between the residents' key opinions and their attitudes towards grazing animals was also examined by using Spearman's rank-order correlation test.

Human Ethics Approval to conduct the survey was granted by the Lincoln University Research Management Office, Human Ethics Committee, Application No: 2020-36.

**3. Results**

*3.1. Positive Attitudes towards the Grazing Animals*

The survey results showed that 89% of respondents felt extremely positive towards the grazing animals at Cornwall Park (as shown in Table 3). A further 10% reported that their attitudes towards the grazing animals are 'mostly positive', meaning a total of 99% of participants having a positive attitude towards the animals. Only 1% of respondents expressed neutral opinions, and no respondents indicated negative attitudes towards the animals.

As shown in Table 4, the significance of the correlations tested between the residents' attitudes towards grazing animals and their socio-demographic and park visitation attributes ranges from 0.06 to 0.97 ($0.06 < \rho < 0.97$). No significant correlation was evident, indicating that the residents' socio-demographic attributes (i.e., age, household type, and proximity) and park visitation attributes (i.e., importance for residing decision and use frequency) have no significant impacts on their attitudes towards the grazing animals within the park.

**Table 4.** Correlations between the respondents' attitudes towards grazing animals and their socio-demographic and park visitation attributes.

| | Attribute | Correlation Coefficient | Sig. (2-Tailed) | N |
|---|---|---|---|---|
| Socio-demographic attributes | Q1. Age | 0.01 | 0.97 | 73 |
| | Q3 & 4. Ownership | 0.03 | 0.81 | 73 |
| | Q5 & 6. Proximity | −0.12 | 0.33 | 69 |
| Park visitation attributes | Q7. Importance for residing decision | 0.22 | 0.06 | 73 |
| | Q8. Use frequency | 0.19 | 0.12 | 73 |

A Pearson Chi-Square test shows that there is no significant association between the residents' attitudes toward the grazing animals and their household type, $X^2$ (8, N = 73) = 5.3, $p = 0.72$, and a range of park visitation motivations (as shown in Table 5).

**Table 5.** Pearson Chi-Square results showing the relationships between the respondents' attitudes towards grazing animals and their socio-demographic and park visitation attributes.

| | Attribute | Pearson Chi-Square | df | Asymptotic Significance (2-Sided) |
|---|---|---|---|---|
| Socio-demographic attributes | Q2. Household type | 5.31 | 8 | 0.72 |
| Park visitation attributes | Q9. Use: recreation | 0.39 | 2 | 0.83 |
| | Q9. Use: dog walking | 0.82 | 2 | 0.67 |
| | Q9. Use: commuting | 0.81 | 2 | 0.67 |
| | Q9. Use: exercise | 0.81 | 2 | 0.67 |
| | Q9. Use: using park facilities | 1.09 | 2 | 0.58 |
| | Q9. Use: other | 1.26 | 2 | 0.53 |

Similarly, by testing the correlation between the respondents' attitudes and the experiences they had with the animals, this research found that there is no significant correlation between the residents' experience and their attitudes (as shown in Table 6, $0.24 < \rho < 0.92$). This means that the positive or even negative experiences they had with the animals have no significant impact on their attitudes.

**Table 6.** Correlations between the respondents' attitudes towards grazing animals and their self-report experiences with the animals.

| Attribute | Correlation Coefficient | Sig. (2-Tailed) | N |
|---|---|---|---|
| Have only positive experiences | 0.20 | 0.87 | 73 |
| Have positive experiences | 0.01 | 0.92 | 73 |
| Have negative experiences | −0.14 | 0.24 | 73 |

Following the 10 'tick-box' questions that formulated the base-line information from participants, four short answer questions were then asked to elicit personal responses to the following questions:

Please provide a brief explanation of what you like about having grazing animals within the park.

Please provide a brief explanation of what you do not like about having grazing animals within the park.

Please provide a brief description of any positive or negative experiences you have had with the grazing animals (sheep and cattle) within Cornwall Park.

What, if anything, would you change regarding how grazing animals are integrated into the park?

Key points taken from the analysis of the four short-answer questions are presented within the sections below. Quotations from the surveys are presented verbatim, using New Zealand English, with any additional words inserted using square brackets [].

*3.2. Understanding the Positive Perception and Experience of Grazing Animals within the Public Park*

Reasons provided by respondents as to their positive experiences and views of the animals within Cornwall Park are illustrated in Figure 5 below. Note, each 'theme' is documented with both the number of participants referring to that theme, as well as the percentage of respondents (shown as a %).

The most frequently mentioned explanation for positive participant attitudes toward grazing animals was described as there being a sense of country or nature that the animal-based agricultural landscape brings to the urban environment (see Figure 5). Participant response from zone 2a, "I love seeing the wonder and enjoyment they bring children and look forward to one day sharing the experience with our grandchildren. So many city children have no experience of farm animals, so this offers a unique experience and opportunity". It was mentioned by a number of respondents that the grazing animals make them feel that they are away from the busy city and keep them connected to nature.

"Keeping animals in [the] park is [a] way to connect city people with nature. More animals please!" (participant response from zone 1b). Apart from the 'rural feelings', 26% and 15% of respondents are attracted by the farm ambiance and practices, respectively. A small group of respondents also remarked on their emotional connection to the 'rural roots' and how the grazing animals provide them with a sense of comfort and nostalgia.

**Country in city (60, 82%)**
A sense of country or nature in the city (40, 55%)
Good to have a working farm in the city (19, 26%)
Seeing, knowing, or experiencing farm practices (11, 15%)
Emotional connection to the countryside (3, 4%)

"To have a 'living' farm in the centre of a major city is a privilege. It brings the countryside close."

"I like it because it reminds me of my childhood - my family owned a dairy farm."

**Seeing, hearing, smelling, or interacting with the animals (52, 71%)**
Seeing or interacting with animals (34, 47%)
Feel privileged to witness the birth and growth of lives (28, 38%)
General preference of animals (16, 22%)
Sounds of animals (11, 15%)
Smell of animals (1, 1%)

"It was a privilege to watch cows and sheep give birth over winter. The gathered groups of old and young were collectively in awe. "

"Local adults, including myself often pat some of the big cows. Also say good day to them."

"The sounds at night are soothing."

**Good for specific groups (44, 60%)**
Good for children (37, 51%)
Good for city dwellers (14, 19%)
Good for visitors and tourists (9, 12%)

"Children love the baby lambs and foreign visitors love the experiences."

"They are a tourist attraction and make nearby residents feel good when they see animals in a natural environment."

**Adding values (26, 36%)**
Making the park unique and interesting (17, 23%)
Productive land use (8, 11%)
Aesthetic values (5, 7%)
Economic values (5, 7%)
Environmental values (3, 4%)

"Living on a farm in the middle of our largest city is something very special and the animals make it real, without them it seems empty."

"Keeps the land productive ..."

"Gives the park an income and purpose as well as beauty."

"Good for the environment (so long as stock variation and rotation occurs to maintain diversity)."

**Education (15, 21%)**
Educational values (9, 12%)
Good for public awareness (7, 10%)

"Great educational value for all - especially children who would not otherwise experience such things."

"I think it's very important for people in the city to see animals; to see how they live builds deeper connections with importance of agriculture, livestock, their life."

**Animals help with the Land management (11, 15%)**
Keep the grass down (9, 12%)
Reducing the risks of fire hazards (6, 8%)
Reducing vermin (3, 4%)

"They are a natural method of keeping pasture attractive, safe from fire and vermin."

"I am an ex-farmer and like the animals in the park and it keeps the park neat and tidy and the grass under control. Look at Mt Eden. No animals to graze the summit. One day in mid-summer there will be a fire."

**Mental health (6, 8%)**    "Have a calming effect on my senses."    "I find it relaxing to see and hear the animals."

**Others (13, 18%)**

**Figure 5.** Self-reported reasons why the respondents like having grazing animals within the park, including example quotations.

The positive attitudes expressed towards the presence of animals, especially for the new lambs and calves during spring were the second most frequent explanation given for the positive attitudes illustrated by the surveyed residents (see Figure 5). Over one-third

of respondents commented specifically on the enjoyment of witnessing the birth of lambs and calves during spring, and also watching the animals grow. The responses collected elucidated a high level of engagement and connection felt and experienced by urban neighbors with the agricultural landscape within Cornwall Park through the everyday perceptual opportunities to see, hear, interact with, and even smell the animals. participant response from zone 3a, "Grazing animals give 'life' to the park. I enjoy seeing the sheep (and lambs), cattle (and calves), as well as the bird life-chickens, pheasants, magpies, and many others. My family have also enjoyed spotting the rabbits too. There is plenty of room for people too, so the animals do not hinder our movement/use whatsoever".

Some specific groups were frequently mentioned in the responses explaining the aesthetics of the grazing animals. Children, and the opportunities provided to them by being able to see and experience the animals were mentioned by more than half of the respondents (as shown in Figure 5). Respondents commented on the animals as enjoyable elements within the landscape for children, with many specifically commenting on the educational opportunities provided to children to experience nature and the natural lifecycle displayed through the farming cycle. It is also commented by 12% of respondents that the animals have always been a major visitor attraction for the park, offering a unique (rural) experience to urban dwellers participant response from zone 3b, "They are a tourist attraction and make nearby residents feel good when they see animals in a natural environment". Another key reason highlighted by the respondents is that the inclusion of the animals is considered as a value-adding practice of the landscape. As shown in Figure 5, around one-fourth of respondents indicated that the animals make Cornwall Park more unique, interesting, and attractive. Commented by 11% of the respondents, the animal-based urban agriculture landscape was considered as a productive and sustainable way of using the land. Some respondents also commented that the farming practices enhanced the aesthetic, economic, and environmental values of the land. "Creates a form of income to ensure money is reinvested into the maintenance and development of the park and its facilities" (participant response from zone 3b).

Indicated by one-fifth of the respondents, educational values were considered as another key benefit offered by the animals (see Figure 5). It is argued that the farming practices provide great learning opportunities for urban dwellers, especially for city children. participant response from zone 3a, "It is amazing to see the animals all through the year but especially when they are having their babies. Our children get to see the lambs and calves moments after birth and then watch them getting bigger. We also get to talk to them about animal farming for meat/wool which is a useful educational tool". Some others commented that having the general public exposed to the farming practices helps improve their social, cultural, and environmental awareness, as well as increasing the level of agricultural literacy within the urban population.

It was also acknowledged by respondents that the animals also help with the maintenance and management of the landscape, with 15% of respondents commenting on the role grazing animals have in keeping the grass down in a sustainable way, as well as keeping the landscape tidy and attractive. One respondent commented that grazing animals are especially good at keeping the grass controlled on undulating landforms like Cornwall Park, which are difficult to manage using conventional machinery. Six (8%) and three (4%) respondents further indicated, respectively, that while keeping the grass trimmed, the fire risk and vermin control were also well-managed by the grazing. participant response from zone 3b, "They keep the grass length under control where they are allowed to graze, reducing vermin issues and fire risk".

The animals also contribute to the mental health of some residents, commented by 8% of the respondents (see Figure 5). One participant noted that the animals contribute to participant response from zone 3b, "Stress reliever, joy of observing and sharing space, connection with nature, warm fuzzies". They reported that the animals contribute to a therapeutic landscape, describing them as relaxing, calming, stress-relieving, or peaceful. The residents' mental health was benefited through a range of ways, including watching

the animals grazing, interacting with them, or even just hearing the animals mooing and baaing.

In summary, participant responses illustrated a clear appreciation for living close to a farming landscape, where urban residents were able to observe and interact with animals being reported as key drivers of the positive attitudes of the local residents towards the grazing animals. In addition, 59% of respondents also illustrated an understanding of at least one additional benefit other than their 'enjoyment' of the animals and farming landscape, which included environmental benefits, economic benefits, safety benefits (e.g., reducing fire risks), and social benefits.

*3.3. Understanding the Negative Perception and Experience of Grazing Animals within the Park*

The respondents were also asked to explain the factors that they do not like about having grazing animals within the park. Figure 6 illustrates the respondents' comments.

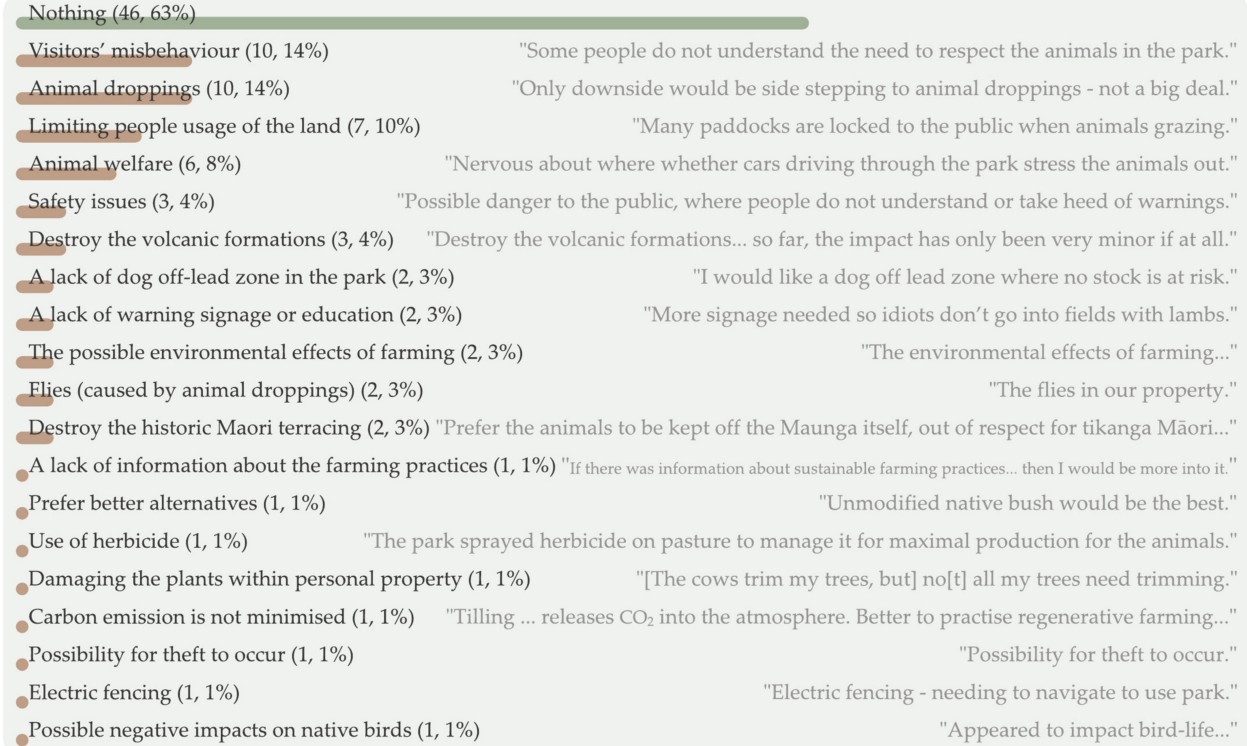

**Figure 6.** Factors reflected by the respondents' answers to the question 'please provide a brief explanation of what you do not like about having grazing animals within the Park'.

As illustrated in Figure 6, the most frequent response to the question 'what do you not like about having grazing animals within the park' was 'nothing', which accounted for almost two-thirds of the responses.

However, for participants who did describe negative views relating to the grazing animals, visitors' misbehavior was the most common response, which was not a reflection on the grazing animals themselves, rather the poor behavior of visitors towards the animals. Responses included details of disrespectful or poorly informed people approaching the animals, chasing them, letting their dogs lose among them, or picking up the young lambs and calves. participant response from zone 2b, "I get annoyed at people who take the park for granted and show little respect for the animals especially mother cows and sheep with their young. They [visitors] try to pick them up, chase them and let their dogs lose among them. It's only a very few but happens every year. It is not a negative about the grazing animals-more the behaviour of humans!". However, many respondents highlighted that

this is indeed not a negative attribute of the grazing animals themselves, rather, it is the misbehavior of humans.

Animal droppings are another issue which was indicated by 10 (14%) respondents as being a point of negativity in relation to the presence of grazing animals within the park. However, most of the comments mentioning animal droppings show a high tolerance and acceptance towards this issue, i.e., they felt that the impacts of animal dropping were minor, and it would be something to be expected. participant response from zone 1b, "Only downside would be side stepping to animal droppings-not a big deal".

Seven (10%) respondents commented that a downside of having grazing animals is that there have to be some limitations on human activities, including road closures (while animals are moved through the park), and limitations on dog walking. One example response illustrating resident frustration to having walking tracks closed during the lambing season stated "Not being able to access the park from our house (when walking) with the dog during lambing season. [We] Have to put [the] dog in [the] car and drive! We live 6 houses away from gate access" (participant response from zone 2a). These limitations happen mainly during the lambing and calving seasons, or when animals are being mustered for shearing. However, some of these respondents remarked that the enjoyment of having the grazing animals far outweighs these limitations. participant response from zone 1a, "Alteration in [my] favourite walks during lambing when fields closed off. However, [I] realise it is important not to disturb the animals and the enjoyment of having the lambs far outweighs the negative". Figure 7 indicates the general areas that are temporarily closed for dog walking during Spring.

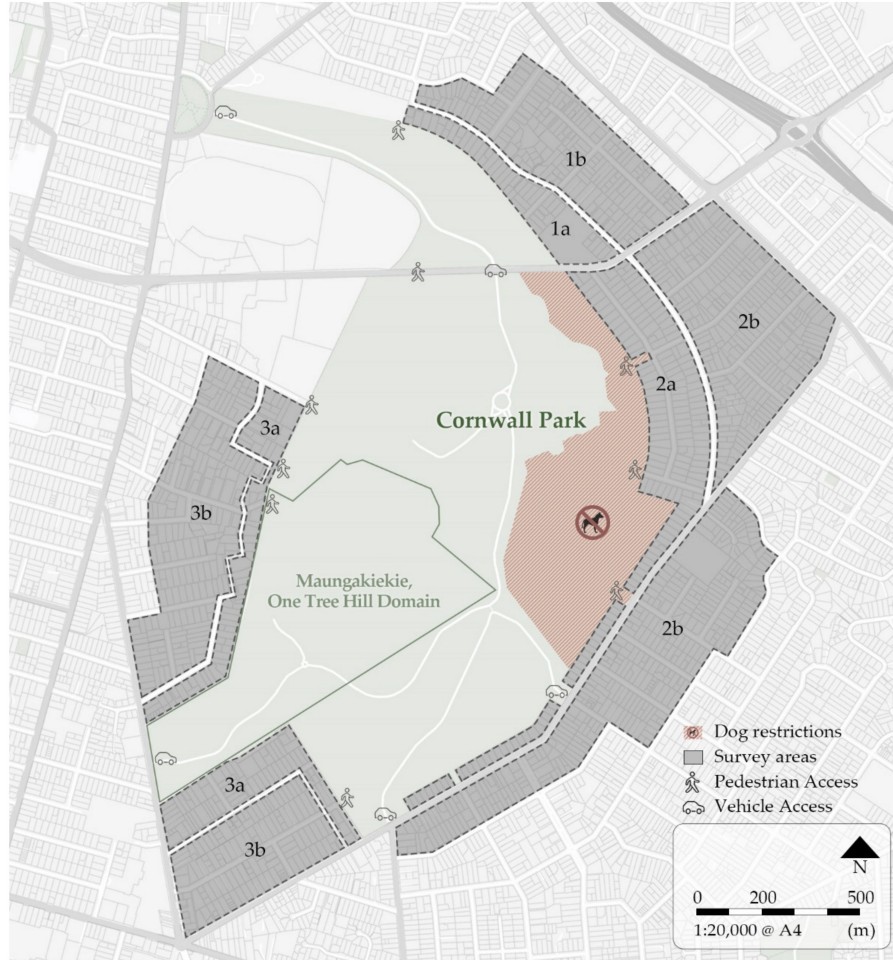

**Figure 7.** "A section of the eastern side of the park will have temporary restrictions applying over the lambing and calving seasons" (Based on information from Edmonds [24]).

Another 8% of respondents expressed concerns about animal welfare. They considered the poorly behaved visitors, their dogs, and the through vehicular traffic as potential risks to the animals, especially in lambing and calving seasons. Many of these respondents also expressed a strong desire for protecting the animals when they were asked for suggested improvements to the integration of grazing animals in a following question. participant response from zone 1b, "I do sometimes worry about the impact on the animals of having such frequent human contact-and not always respectful contact although the park farmers do well of trying to limit this and educate the public".

Other concerns respondents commented on in relation to the negative impact of grazing animals within the park included, a lack of dog off-lead zones; the potential damaging impacts of the animals on the volcanic formations found within the park landscape; the negative impact of animals on the protection of historic Māori landscape features present within the park (e.g., historical terracing associated with fortification and food production and storage); risks associated directly with the animals themselves (e.g., aggressive behavior); and finally the possible environmental effects of unsustainable farming practices. participant response from zone 2a, "The park sprayed herbicide on pasture to manage it for maximal [maximum] production for the animals. Commercial farming isn't done for the benefit of the animals or the environment. I would prefer to live next to a park that was managed for its biodiversity values/ecosystem function. Unmodified native bush would be the best".

Nevertheless, it is noteworthy that the majority of comments explaining respondents' concerns with having animals within the park still had no objections to the inclusion of grazing animals. On the contrary, most of them appreciate the existence of the animals, showing a strong desire within their comments to support the grazing animals and farming practices. One participant responded participant response from zone 2b, " . . . add more [animals]! More diversity, more breeds, and different animals. I would try and grow certainly more crops in there for food and change each field over with new stock and feed to introduce more diversity".

The final short answer question of the survey asked participants to comment on 'what, if anything, would you change regarding how grazing animals are integrated into the park?'. Noted by 59% of participants, the response to this question was 'change nothing', with one participant stating "I think the park staff do an absolutely fabulous job with the farm and the grazing animals. We are so grateful to have such a beautiful natural asset on our doorstep and we feel it enriches our lives and our experience of living in Auckland, every single day" (participant response from zone 3b).

Additionally, 18% of respondents explicitly mention a desire for more animals (noting specifically a desire to include a greater variety of animals, and to also include smaller animals), and more space within the park for grazing. Nineteen percent of responses noted a desire for more information and educational opportunities in relation to the farm and animals. Another 4% of respondents commented on a desire to see organic farm practices, regenerative cropping, alternative animal management systems or a diversification of land use. Respondents also felt positively towards the park and farm management, with one participant stating. participant response from zone 2b, "I think the farm manager (does) and workers do a superb job of looking after the fields and animals through all the seasons. I wouldn't change anything-they are the experts. Perhaps the area around the shearing shed could have chickens and pigs and become a working farm for schools/children/tourists to have tours-see the dogs working, etc.

## 4. Discussion

This case study has illustrated neighbor perspectives of the integration of animal-based urban agriculture within an urban greenspace of Tāmaki Mataurau Auckland, Aotearoa New Zealand's largest urban center. Through surveying residents who reside in homes located close to Cornwall Park, the aim of this research was to better understand the enablers and barriers to integrating livestock animals within urban greenspace from a 'neighbor'

perspective. The survey results have clearly illustrated the affection and appreciation residents have for the animals, and the perceived role they play in their lives and the lives of their children, family, and visitors alike. participant response from zone 1b, "We love spring when the animals are calving etc. and feel privileged that our children have grown up with such easy access to this aspect of nature".

Findings revealed that the grazing animals were highly valued by the neighboring community with 99% of respondents feeling 'positive' (including responses as 'mostly positive' and 'extremely positive') towards the inclusion of grazing animals as part of the public park. Residents felt a sense of 'privilege' to be able to live near a park that included a working farm and felt a strong sense of guardianship towards the animals. participant response from zone 3b, "It feels such a privilege to live in a city which has a real working farm at its heart, and we purchased out property adjoining C.P. [Cornwall Park] solely because of its proximity to the park. We hear the animals from our home and see them on a daily basis-it feels like a little piece of country paradise in the centre of the city".

This study has illustrated the idea that grazing animals are a favorable element that can be well-integrated into urban parks, with the overwhelming majority of residents enjoying having grazing animals within the urban setting. This research has indicated that animal-based urban agriculture is highly favorable to the majority of park neighbors and could potentially be integrated into urban settings with different demographic or geographic contexts, given that age, home ownership status, proximity to the park, or reasons for using the park, showed no significant correlation with attitudes towards the animals. The weak correlations, in turn, highlight that reintegrating animals into urban environments is highly transferable and adaptable for future practices in different settings, as the positive attitudes are predominantly driven by biophilic effects, that are shared by the majority, rather than the socio-demographic and park visitation attributes of individuals, which are different from one to another.

Perceived negative aspects of animal integration that may indicate barriers to the integration of animals within urban agriculture systems, included issues associated with animal welfare (primarily due to human factors such as misbehavior and poor dog control), animal droppings, and the seasonal limitations the animals put on publicly accessible space within the Park, particularly during springtime when the public is asked to stay out of fields where sheep and cattle are lambing and calving. As indicated by the respondents themselves, in many of the explanations around negative perceptions and experience of the grazing animals, was the notion that the issues highlighted by many participants were seen as being 'human-induced', participant response from zone 2b, "I would rather see a change to how visitors behave with the animals". Other factors such as creating mild inconvenience, were often then identified as a necessary 'inconvenience' where the 'greater good' of being able to include the animals in the park was acknowledged by many respondents.

Overall, the responses received through the survey illustrate a significant positive attitude towards the animals and the farming systems present within Cornwall Park. The surveyed community who live around Cornwall Park feel 'privileged' to live near a working farm that they can access, observe and interact with the animals. When asked to discuss possible changes for the future of the park, 56% of participants noted that they would not change anything, with an additional 14% indicating a desire to expand the grazing areas and increase the number and type of animals, with one participant stating. Participant response from zone 2a, "Expand the grazing area and perhaps introduce goats, horses, and more varieties of sheep".

## 5. Conclusions

These findings have implications and meaning for cities of the Global North considering the reintegration of animal-based urban agriculture. Providing qualitative evidence by way of resident surveys, this research aimed to better understand the enablers and barriers to integrating livestock animals within urban greenspace from a 'neighbor' perspective.

This paper presents survey responses from residents living around Cornwall Park, an urban greenspace in Aotearoa New Zealand's largest city, Tāmaki Makaurau Auckland, that integrates a working sheep and beef farm as part of the 172ha urban greenspace. Findings revealed that the grazing animals were highly valued by the neighboring community with 99% of respondents feeling 'positive' towards the inclusion of grazing animals as part of the public park.

Findings identified a number of key issues that defined community perception and feelings of positivity towards including grazing animals within the park. These findings indicate design and policy implications for future planning decisions for the re-integration of animal-based urban agriculture. Firstly, the community surrounding the park, on the whole, felt positively towards the animals and having them living within the park. However, there were concerns about animal welfare, mainly due to a perceived lack of understanding of visitors to the park with regard to how to interact respectfully with the animals. This issue could be mitigated (as suggested by some participants) by using enhanced educational and informational techniques within the park. These might include interpretation boards, signage, open days, and educational programs for visitors.

Secondly, the community surrounding the park, on the whole, feels privileged to live near a working farm, and have the opportunity to experience all that this brings to their life within an urban setting. However, there were concerns highlighted around the environmental impact of 'conventional' agricultural practices, and there was a desire noted by some participants suggesting the park investigate organic, regenerative, and more diverse farming systems, as a way to further protect the land, the archeology, and the neighbors from possible negative effects, e.g., from spraying and the use of fertilizers. This issue could be positively addressed in the initial design, planning and public consultation for animal-based urban agriculture, where alternative farming practices, acceptable and informed by public input could be developed in-line with urban aspirations. With this said however, the majority of participants within this research highlighted the excellent work farm management and staff do, sometimes within very trying situations when dealing with the public, due to the highly visible and accessible nature of the park and the animals, and a 'perceived' lack of agricultural literacy from visitors.

Thirdly, this research highlighted the positive perception that grazing animals bring to the fire safety of the park, by way of low-input grass maintenance. This positive environmental aspect of urban grazing could further enhance policy and acceptance for animal-based urban agriculture.

Understanding these key issues illustrated by this research can provide guidance and support for decision-making when defining policies for enabling animal-based agriculture within public greenspace. This investigation shows a clear and definitive desire by residents living close to an urban farm within a public greenspace to continue integrating grazing animals within the park, with the research highlighting a broad range of insights from a community perspective. Understanding the negative issues and experiences of integrating animal-based urban agriculture within cities highlighted by this case study, provides important insight and strategic opportunity for the future planning and design of animal-based urban agricultural systems within urban environments of the Global North that pre-empt and are designed to mitigate possible negative impacts.

The research presented within this paper has focused on community perceptions of animal-based urban agriculture, highlighting the positive and negative attitudes and experiences of residents living near Cornwall Park as a case study. Further research into the complex and sometimes conflicting nature of animal-based urban agriculture is required as the Global North reconsiders the role agriculture and productive landscapes have within urban environments, as key conversations around urban resiliency and sustainability continue. Issues associated with animal welfare, environmental impacts, and the transmission of disease are all aspects that require further investigation.

For the past 100 years city authorities throughout the Global North have actively planned animals out of cities, however this research highlights the range and breadth

of positive opportunity that animals contribute to people, communities and the urban environments. By understanding the key issues highlighted above, decisions makers, planners and urban designers within the Global North can begin planning for the successful re-integration of animals into the urban realm, in this case within public parks.

**Author Contributions:** Conceptualization, S.D.; methodology, S.D.; data analysis, G.C.; investigation, S.D. and G.C.; data curation, S.D. and G.C.; writing—original draft preparation, S.D. and G.C.; writing—review and editing, S.D. and G.C.; funding acquisition, S.D. All authors have read and agreed to the published version of the manuscript.

**Funding:** This research was funded by the Faculty of Environment, Society and Design, Lincoln University, New Zealand and the Lincoln University Open Access Fund.

**Institutional Review Board Statement:** This research was approved by the Lincoln University Human Ethics Committee, HEC2020-36.

**Informed Consent Statement:** Informed written consent was obtained from all subjects involved in the study.

**Data Availability Statement:** Restrictions apply to the availability of these data. Anonymised data was obtained from surveyed residents of Auckland and may be available from the authors with the permission of the Lincoln University Human Ethics Committee.

**Acknowledgments:** The authors would like to acknowledge the support of Jacky Bowring for providing feedback on the final manuscript.

**Conflicts of Interest:** The authors declare no conflict of interest. The funders had no role in the design of the study; in the collection, analyses, or interpretation of data; in the writing of the manuscript, or in the decision to publish the results.

## Glossary

| | |
|---|---|
| Māori | indigenous people of New Zealand |
| Maunga | mountain |
| Tikanga | customary system of values and practices |

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
