# Peer review of "Community Perception of Animal-Based Urban Agriculture within City Greenspaces of the Global North: A Survey of Residents near Cornwall Park, New Zealand"

_sustainability, doi:10.3390/su141912419_

Round 1
Reviewer 1 Report
Excellent work presented in the article. The study focused on understanding barriers for re-integrating grazing animals within cities of the Global North; focusing on community perceptions toward the inclusion of a sheep and cattle farm within an urban park. Methodology, results, discussion and conclusion sections are well presented. However, need to discuss more about animal welfare aspects and include educational information related to human health focusing on disease transmission and environmental health.
Author Response
Thank you very much for your review of our paper. We have responded to your points in the attached document. Thank you once again.

Reviewer 2 Report
The entitled ms “Community perception of animal-based Urban Agriculture within city Grenspaces of the Global North A Survey of residents near Cornwall Park in New Zealand” presented the fundamental idea that grazing animals are favourable elements that can be well integrated into urban parks and show no inter-animal effects.
some Positive Comments to the Authors are set out below:
Line 89: rewrite as Specht et al. [21]
On the Methodology, the research design and hypotheses are not clearly stated, thus
Line 384 : correct the number in bracket Seven (7%)
Unfortunately, however, the disadvantages and difficulties that arise become visible.
The question arises, how this result translates into practice? Can you please explain briefly in conclusions?
Author Response

(The authors gave the same response as above.)

Reviewer 3 Report
The idea of gazing at the city areas is an interesting approach.
I have two questions.
(1) In Table 3-5, the correlation is weak. There are many ways to understand the data. Is there any way to analyze the data? Or did you perform an interview with some persons?
(2) In section 3.3, there is a table of negative points. I wonder were there any opinions about the smell of animals and animal droppings?
Author Response

(The authors gave the same response as above.)
